# LATTE FLOWS: LATENT TEMPORAL FLOWS FOR MULTIVARIATE SEQUENCE ANALYSIS

## ABSTRACT

We introduce Latent Temporal Flows (*LatTe-Flows*), a method for probabilistic multivariate time-series analysis tailored for high dimensional systems whose temporal dynamics are driven by variations in a lower-dimensional discriminative subspace. We perform indirect learning from hidden traits of observed sequences by assuming that the random vector representing the data is generated from an unobserved low-dimensional latent vector. *LatTe-Flows* jointly learns auto-encoder mappings to a latent space and learns the temporal distribution of lower-dimensional embeddings of input sequences. Since encoder networks retain only the essential information to generate a latent manifold, the temporal distribution transitions can be more efficiently uncovered by time conditioned Normalizing Flows. The learned latent effects can then be directly transferred into the observed space through the decoder network. We demonstrate that the proposed method significantly outperforms the state-of-the-art on multi-step forecasting benchmarks, while enjoying reduced computational complexity on several real-world datasets. We apply *LatTe-Flows* to a challenging sensor-signal forecasting task, using multivariate time-series measurements collected by wearable devices, an increasingly relevant health application.

## 1 INTRODUCTION

One of the core objectives in machine learning research is to build models that accurately capture and explain complex structures in real-world systems. Much of unsupervised learning is driven by probabilistic generative modeling, which is notoriously difficult in high dimensions as a consequence of "the curse of dimensionality." For example, biomarker observations (e.g. heart rate and respiratory rate) collected from wrist-worn devices are naturally collected sequentially from a large number of correlated features.

Patterns and complex interactions in sensor signals have been shown to be associated with changes in physiology (Goodale et al., 2019). Recorded time-series from different sensors can form multivariate time-series data. Joint probability modeling of these multivariate sensors across time allows for analysis of expected or abnormal patterns in physiological vital signs. Expressive time-series generative models can be used to perform sequential data analysis and forecasting. Specifically, these models allow for long-term predictions with uncertainty estimation (Futoma et al., 2016), time-series data augmentation (Wen et al., 2020), out-of-distribution detection (Serrà et al., 2019), missing data imputation (Luo et al., 2018), and improved model interpretability analyses (Ismail et al., 2020; Rooke et al.). Such information can be integrated into health monitoring systems (Dunn et al., 2021).

Given a sequence of data of dimensionality $N$, with $N$ often large, a multivariate time-series forecasting model aims to effectively predict the future outcomes of each variable $n \in \{1, \ldots, N\}$, given their past values. Typically, this process requires an accurate estimate of a multivariate conditional distribution of temporal transitions, making the task challenging in the high-dimensional setting. This task is challenging in terms of computational complexity and expressiveness due to the number of parameters required to be estimated for the multivariate parameterization. Thus, directly estimating the full predictive distribution of the future values in the high-dimensional observation space is sample-inefficient and may require a large amount of training data. The intractability of estimating such models for large systems, due to the growth in the number of parameters with di-

mension, has limited existing methods to handle at most a few dimensions or requires restrictive assumptions such as tractable distribution classes (Chung et al., 2015) or low-rank approximations (Salinas et al., 2019a). An alternative way to tackle time-series generative modeling is to view the collection of data as $N$ separate time sequences and fit a separate model for each of the dimensions. However, univariate techniques (Zhang et al., 2017; Oreshkin et al., 2019; Montero-Manso et al., 2020), do not benefit from joint leaning of temporal dependencies between features and are thus limited in forecasting accuracy. Since many multivariate time-series in practical applications are highly correlated, it is crucial to learn both intra-series and inter-series patterns. Modeling dependency relationships among the variables between the individual time-series, including indirect relationships through shared latent causes (Wang et al., 2019), shows promise in enhancing the performance of forecasting.

Data dependencies may be hidden in high dimensions due to noise and high variance, and thus can be missed when conducting direct data analysis in the higher dimensional space. In such cases, we aim to learn dynamics in a compact latent space to enable faster forecasting, and potentially reveal latent trends. We assume that the observed, possibly high-dimensional, random vector representing the data of interest, is generated from an unobserved low-dimensional latent vector through a time-varying probabilistic process (Zhang et al., 2019), (Laumer et al., 2020), (Louis et al., 2019), (Chung et al., 2015), (Yoon et al., 2019), (Nguyen & Quanz, 2021).

By modeling a lower-dimensional latent vector by a (stationary) reversible embedding mapping, *LatTe-Flows* increases parsimony, while reducing computational complexity. These mappings are trained to favor latent representations that uncover latent dynamics over future latent states. Under *LatTe-Flows*, forecasts are generated by estimating the latent state evolution, along with recovery functions. Nonlinear mappings are trained via deep auto-encoder networks, while latent transition dynamics are learned using a combined structure of Normalizing Flows (Rasul et al., 2020; Dinh et al., 2016; Papamakarios et al., 2017) and a multivariate Recurrent neural network (RNN) (Graves, 2013; Sutskever et al., 2014).

Our extensive experiments on real-world datasets illustrate *LatTe-Flows*' state-of-the-art performance and computational tractability on the majority of the datasets considered. To demonstrate the ability of the proposed model to match the ground truth trajectory distribution, we apply *LatTe-Flows* on the Apple Heart and Movement Study (AH&MS) dataset for a challenging vital sensor-signal forecasting task. We visualize the 2-dimensional and 3-dimensional learned latent representations from our model to showcase the interpretable low-dimensional embeddings. These emebeddings show that *LatTe-Flows* is able to easily identify participants' $VO2_{max}$, a main indicator and summary of cardiorespiratory fitness, while only being trained on relatively lower-level vital signals like resting heart rate and heart rate variability.

## 2    RELATED WORK AND OUR CONTRIBUTIONS

The literature on time-series forecasting has a long history—in this work, we mainly focus on recent developments in the deep learning context. Simple models for multivariate data, such as general State-Space Models (SSMs) (Hamilton, 2020; Liu et al., 2016), N-BEATS (Oreshkin et al., 2019), Gaussian Processes (Rasmussen, 2003), DeepAR (Salinas et al., 2020; Zhang et al., 2017) learn one model per time-series (univariate methods). As a consequence, they cannot effectively capture complex structure and interdependencies between multiple time-series.

Although multivariate probabilistic time-series forecasting models estimate the full predictive distribution, the number of parameters in these models grows quickly with the number of variables, which results in large computational cost and a high risk of overfitting in high dimensional settings. Methods such as Variational Recurrent Neural Networks (VRNNs) (Chung et al., 2015) or Time-GAN (Yoon et al., 2019), either assume pre-selected tractable distribution classes or another type of structural approximation (Salinas et al., 2019a). In the low-rank Gaussian copula model, for instance (Salinas et al., 2019a), a multitask univariate LSTM (Hochreiter & Schmidhuber, 1997b) is used to output transformed time-series and diagonal and low-rank factors of a Gaussian covariance matrix. These assumptions can limit the distributional expressiveness of low-rank Gaussian copula models.

Recently, more flexible models such as Temporal Conditioned Normalizing Flows (Rasul et al., 2020) have been proposed. This method uses a multivariate RNN to learn temporal dynamics with the state translated to the output joint distribution via a Normalizing Flow (Dinh et al., 2016). However, during forecasting, an invertible flow is applied on the same number of latent dimensions as input dimensions, thus it does not scale to large numbers of time-series (since RNNs have quadratic complexity in $N$). The work most closely related to ours is TLAE (Nguyen & Quanz, 2021), where the temporal model is applied across a low dimensional space. This method combines an RNN-based model with auto-encoders to learn a temporal deep learning latent space forecast model. TLAE focuses on the encoder/decoder modeling capabilities and assumes simplistic probabilistic structure on the latent vector (multivariate Gaussian with diagonal covariance matrix), which can be restrictive. We stress that the encoder mapping for TLAE is time-dependent, along with the probabilistic model.

Our key contributions in this work can be summarized as follows:

- **LatTe-Flows can model cross-series dependencies across time while scaling to large dimensions.** We capture the joint temporal cross-series dynamics, by pairing non-linear dimensionality reduction with latent temporal distribution estimation, while scaling to a large number of time-series in $\mathbb{R}^N$ as a consequence of reduced dimensionality of the (joint model) learning space.

- **Forecasting in LatTe-Flows is performed across a low dimensional space, enabling faster sequence generation.** Instead of applying the temporal model across all series, multivariate forecasts in *LatTe-Flows* are generated across a low dimensional space, and then mapped back to the observation space.

- **We allow flexibly and expressive probabilistic models in the latent space.** To increase model flexibility, we introduce explicit learning of the latent conditional distribution of temporal transitions without strong assumptions of traditional multivariate models.

- **We introduce a flexible end-to-end training process.** We are able to harness stochastic gradient descent by combining the objectives for sequence reconstruction and latent probabilistic prediction.

## 3 PROBLEM STATEMENT AND PROPOSED APPROACH

Consider a collection of high dimensional multivariate time-series $\mathbf{y}_{n,t} \in \mathbb{R}$, where $n \in \{1, 2, \ldots, N\}$ indexes the individual univariate component time-series, and $t$ indexes time. Consequently, the multivariate observation vector at time $t$ is given by $\mathbf{y}_t \in \mathbb{R}^N$. Given a sequence of $T_{\text{total}}$ vector realizations of $\mathbf{y}_t$, a multivariate time-series can be represented as a matrix $\mathbf{Y}$, $\mathbf{Y} \in \mathbb{R}^{N \times T_{\text{total}}}$.

We focus on the task of multivariate time-series multi-step forecasting. More formally, let us assume that we are given an observed history $(\mathbf{y}_1, \ldots, \mathbf{y}_T)$, sampled from the complete time-series history of the training data, where each instance consists of $N$ temporal features (that occur over time, e.g. vital sensor signals). Our goal is to learn future values of the series over a length-$\tau$, $\tau > 0$, forecast horizon and predict a set of plausible future trajectories $(\hat{\mathbf{y}}_{T+1}, \ldots, \hat{\mathbf{y}}_{T+\tau})$ by learning the conditional distribution $p(\mathbf{y}_{T+1:T+\tau}|\mathbf{y}_{1:T})$ of temporal transitions. We refer to time-series $\mathbf{y}_{n,1:T+\tau}$ as the target time-series, which for training is split according to a time range $(1, 2, \ldots, T)$ referred to as context window, and to time $(T+1, T+2, \ldots, T+\tau)$ as prediction horizon.

The temporal dynamics of complex systems are often driven by fewer and lower-dimensional factors of variation (Laumer et al., 2020), (Louis et al., 2019), (Chung et al., 2015), (Yoon et al., 2019), (Nguyen & Quanz, 2021), (Amiridi et al., 2021). We assume that the observed, possibly high-dimensional, random vector $\mathbf{y}_t \in \mathbb{R}^N$ representing the data of interest is generated from an unobserved low-dimensional latent vector $\mathbf{x}_t \in \mathbb{R}^D$ with $D \ll N$ through a time varying probabilistic process $p(\mathbf{x}_t|\mathbf{x}_{1:t-1})$. To reduce the computational burden while at the same time improving distribution modeling, we propose incorporating representation learning in the generative learning problem to explicitly learn the temporal distribution of compact representations of input sequences. Specifically, *LatTe-Flows* consists of three key components: an embedding function $g : \mathbb{R}^N \to \mathbb{R}^D$, a latent conditional distribution $p(\mathbf{x}_t|\mathbf{x}_{1:t-1})$ of temporal transitions, and a recovery function $q : \mathbb{R}^D \to \mathbb{R}^N$. During training, the model simultaneously learns to produce sequence

representations that will push a latent temporal generative model to predict plausible latent future sequences, recover them back into the observed space, and iterate across time.

Given a model for $p(\mathbf{y}_{T+1:T+\tau}|\mathbf{y}_{1:T})$, one can estimate the conditional expectation, which can be expressed as a function of past observations: $\mathbb{E}[\mathbf{y}_{T+1:T+\tau}|\mathbf{y}_{1:T}] = f(\mathbf{y}_1, \ldots, \mathbf{y}_T)$. An indirect estimate of this function can be accomplished using the following strategy: during testing, future trajectories $(\hat{\mathbf{y}}_{T+1}, \ldots, \hat{\mathbf{y}}_{T+\tau})$ are generated by embedding the past history via $g : (\mathbf{x}_1 = g(\mathbf{y}_1), \ldots, \mathbf{x}_T = g(\mathbf{y}_T))$, sampling from the latent future distribution $(\hat{\mathbf{x}}_{T+1}, \ldots, \hat{\mathbf{x}}_{T+\tau}) \sim p(\cdot|\mathbf{x}_{1:T})$, followed by applying a non-linear recovery function $q$: $(\hat{\mathbf{y}}_{T+1} = q(\hat{\mathbf{x}}_{T+1}), \ldots, \hat{\mathbf{y}}_{T+\tau}\} = q(\hat{\mathbf{x}}_{T+\tau}))$. Next, we describe each building block of our approach and the combined training strategy.

## 3.1 Dimensionality Reduction of High-Dimensional Sequences

**Learning Sequence Representations**

The goal of the proposed method is to obtain suitable representations $\mathbf{x}_t \in \mathbb{R}^D$ that reveal a reduced search space for future sequence forecasting, where underlying patterns and meaningful information among features is preserved. Latent-space forecasting has been considered both in (Yu et al., 2016) as a result of matrix factorization as well as in (Nguyen & Quanz, 2021). In Yu et al. (2016), a multivariate time-series $\mathbf{Y}$ is decomposed into components $\mathbf{Q} \in \mathbb{R}^{N \times D}$ and $\mathbf{X} \in \mathbb{R}^{D \times T_{\text{total}}}$, with $\mathbf{X}$ temporally constrained. Let us denote $\mathbf{G}$ as the pseudo-inverse of $\mathbf{Q}$. If $\mathbf{Y}$ can be decomposed by $\mathbf{G}$ and $\mathbf{X}$, forecasting for the high-dimensional series $\mathbf{Y}$ can be performed on a smaller dimensional series. In TLAE (Nguyen & Quanz, 2021), linear mappings are generalized to nonlinear transformations via time-varying auto-encoders. In our framework, each input vector $\mathbf{y}_t \in \mathbb{R}^N$ is mapped to a condensed representation $\mathbf{x}_t \in \mathbb{R}^D$ (usually $D \ll N$) using a *stationary* reversible embedding mapping $g : \mathbb{R}^N \to \mathbb{R}^D$. Such representations are trained to reveal a low-dimensional structure, which allows an expressive family of temporal constrained distributions to fully uncover this, i.e., mappings that yield a low negative log-likelihood cost over $p(\mathbf{x}_t|\mathbf{x}_{1:t-1})$.

The first component of our model is estimating the dimensionality reducing mapping. Exploiting the deep neural network's ability to to extract higher order features (Yang et al., 2017) and approximate any nonlinear function, we replace $\mathbf{G}$ by an encoder and $\mathbf{Q}$ by a decoder neural network. An encoder network $g_\phi : \mathbb{R}^N \to \mathbb{R}^D$, embeds $\mathbf{y}_t$ at time $t$ into a low-dimensional latent space, vector $\mathbf{x}_t$, using a nonlinear map. Operating within the latent space, we then seek, a dynamical system that prescribes a rule to move forward in time, and a decoder network $q_{\phi'} : \mathbb{R}^D :\to \mathbb{R}^N$, $\hat{\mathbf{y}}_t = q_{\phi'}(\mathbf{x}_t)$ to reconstruct latent variables in the spatial domain. Although the embedding mapping is assumed to be stationary, latent representations' progressions over time are captured via an autoregressive deep learning model, where the data distribution is represented by a conditioned Normalizing Flow. During training, the auto-encoder learns by fine-tuning the parameters of a feed-forward Deep Neural Network (DNN) in such a way that the reconstruction error is minimized when back projected with another feed-forward DNN. These networks need to be specified a-priori, in terms of the number of layers and neurons. We note that the specific requirements for $g$ and $q$ are problem dependent, and we detail the particular design we use in the Appendix. Given a batch of time-series $\mathcal{B}$, and with $|\mathcal{B}|$ denoting the cardinality of the batch set, the first term of our overall cost function consists of the reconstruction loss: $\mathcal{L}_{\text{REC}} := \frac{1}{|\mathcal{B}|(T+\tau)} \sum_{\mathbf{y}_{1:T+\tau} \in \mathcal{B}} \sum_{t=1}^{T+\tau} ||\mathbf{y}_t - q(g(\mathbf{y}_t; \phi); \phi')||_2^2$.

## 3.2 Latent Sequence Modeling

The latent random vector lies at the heart of our overall probabilistic model: It is assumed to "encode" the observed data in a compact manner through $\mathbf{x}_t = g_\phi(\mathbf{y}_t)$, allowing an accurate model of the probabilistic model $p_\theta(\mathbf{x}_t|\mathbf{x}_{1:t-1})$, from which new data can be generated. Our goal is to recover the latent space structure through a flexible time-conditioned distribution model in which the most important features are kept. We also learn the mapping $\hat{\mathbf{y}}_t = q_{\phi'}(\mathbf{x}_t)$ that translates the latent effects to the original data space.

Using the chain rule, the joint distribution of predicted values conditioned on observed values, $p(\mathbf{y}_{T+1}, \ldots, \mathbf{y}_{T+\tau}|\mathbf{y}_{1:T})$, can be written as a product of conditional distributions. Autoregressive models use a neural network to approximate the conditional distribution $p(\mathbf{y}_t|\mathbf{y}_{1:t-1})$ by a paramet-

ric distribution $p_\theta(\mathbf{y}_t|\mathbf{y}_{1:t-1})$ specified by learnable parameters $\theta$. The prediction at time $t$ is input to the model to predict the value at time $t + 1$: $p(\mathbf{y}_{T+1}, \ldots, \mathbf{y}_{T+\tau}|\mathbf{y}_{1:T}) = \prod_{t=T+1}^{T+\tau} p(\mathbf{y}_t|\mathbf{y}_{1:t-1})$. We wish to replace this decomposition by a tractable, approximate statistic $\mathbf{h}_t$ of the past. To represent the history of observations in a compressed state vector $\mathbf{h}_t$ we use RNNs (Graves, 2013; Sutskever et al., 2014), with the most well-known variants, the LSTM (Hochreiter & Schmidhuber, 1997a) and GRU (Chung et al., 2014).

We assume that a time-related (and recursively updated) vector $\mathbf{h}_t \in \mathbb{R}^H$, can summarize the history of the time-series up to time time point $t$, $\mathbf{h}_t = \text{RNN}_\theta(\mathbf{y}_{t-1}, \mathbf{h}_{t-1})$, where $\text{RNN}_\theta$ is a multi-layer LSTM or GRU parameterized by shared weights $\theta$ and $\mathbf{h}_0 = \mathbf{0}$. The state is compressed as it uses less space than the history of observations. Under this model, we can factorize the joint distribution of the observations as $(\mathbf{y}_{T+1}, \ldots, \mathbf{y}_{T+\tau}|\mathbf{y}_{1:T}) = \prod_{t=T+1}^{T+\tau} p_\theta(\mathbf{y}_t|\mathbf{h}_t)$, where now $\theta$ comprises both the weights of the RNN as well as the probabilistic model. This model is auto-regressive as it consumes the observations at the time step $t - 1$ as input to learn the distribution of the next time step $t$. Then, time conditioning on the latent generative model $p_\theta(\mathbf{x}_t|\mathbf{h}_t)$ can be realized by employing a multivariate RNN to model the series progressions, with the state translated to the output joint distribution via a flow (we focus on Real-NVP, but MAF is also explored in our experiments; background on Normalizing Flows can be found in the Appendix ). This combination retains the power of autoregressive models—such as good performance in extrapolation into the future—with the flexibility of flows as an expressive distribution model.

Following Rasul et al. (2020), we concatenate $\mathbf{h}$ to the inputs of the scaling and translation function approximators of the coupling layers, i.e. $s(\text{concat}(\mathbf{x}^{1:d}, \mathbf{h}))$ and $t(\text{concat}(\mathbf{x}^{1:d}, \mathbf{h}))$. The model, which is parameterized by both the flow (the weights of the scaling and translation neural networks) and the RNN $-\theta$ is trained by minimizing $\mathcal{L}_{\text{NegLL}} := \frac{1}{|\mathcal{B}|(T+\tau)} \sum_{\mathbf{y}_{1:T+\tau} \in \mathcal{B}} \sum_{t=T+1}^{T+\tau} \log p(\mathbf{y}_t|\mathbf{h}_t; \theta)$.

## 3.3 THE TRAINING AND TESTING PROCESS

A schematic overview of the training procedure is depicted in Figure 1. Each time-series $\mathbf{Y}$ is split into a training $\mathbf{Y}_{\text{TR}}$ and test set $\mathbf{Y}_{\text{TS}}$ by using all data prior to a fixed date for training and using rolling windows for the test set. A batch input $\mathbf{Y}_B \in \mathbb{R}^{D \times (T+\tau)}$ defines a sub-matrix of $\mathbf{Y}_{\text{TR}}$ with column indices defined by the set $\mathcal{B}$ – it contains two components: the first part $\{\mathbf{y}_t\}_{t=1}^T$ is associated with the past input, while the second component $\{\mathbf{y}_t\}_{t=T+1}^{T+\tau}$ is associated with future observations. A batch of time-series $\mathbf{Y}_B$ is embedded into latent variables via $g : \mathbf{x}_t = g(\mathbf{y}_t)$, yielding a matrix of latent codes $\mathbf{X}_B \in \mathbb{R}^{D \times (T+\tau)}$. To discover informative representations in a lower-dimensional space, each vector $\mathbf{x}_t$ is passed through the decoder layers: $\hat{\mathbf{y}}_t = q(g(\mathbf{y}_t))$, yielding a reconstructed matrix $\hat{\mathbf{Y}}_B$. By minimizing the reconstruction error $\|\hat{\mathbf{y}}_t - \mathbf{y}_t\|_2^2$, the model is expected to capture feature dependencies across time-series and encode this global information into a few latent variables in $\mathbf{x}_t$. Simultaneously, by maximizing the log-likelihood of (latent) future observations given compressed past input, the model targets to capture latent series progressions via time-conditioned Normalizing Flows. The proposed model is meaningful as it encourages the latent variables to capture different complex patterns of the data, which makes the representation more powerful and universal.

The key insight is that the encoding/decoding components and the time conditioned latent temporal generative model are jointly trained by minimizing a combined loss function over a given batch $\mathcal{B}$ of time-series consisting of a reconstruction loss-related term and a sequence negative log-likelihood term using stochastic gradient descent-based optimization. The overall loss term is

$$\mathcal{L}(\phi, \phi', \theta) := \frac{1}{|\mathcal{B}|(T+\tau)} \sum_{\mathbf{y}_{1:T+\tau} \in \mathcal{B}} \sum_{t=1}^T \Big( \underbrace{\|\mathbf{y}_t - q(g(\mathbf{y}_t; \phi); \phi')\|_2^2}_{\mathcal{L}_{\text{REC}}} - \lambda \underbrace{\log p(q(g(\mathbf{y}_t; \phi); \phi')|\mathbf{h}_t; \theta)}_{\mathcal{L}_{\text{NegLL}}} \Big),$$

(1)

By minimizing a combined error of two tractable losses $\mathcal{L}_{\text{REC}}$ and $\mathcal{L}_{\text{NegLL}}$, the model is given the capability to predict the future from latent representations of the observed history that preserve only the essential information, which is transferred to the decoder by minimizing the reconstruction loss. At the same time, it serves to reduce the dimensions of the temporal generative learning space. The

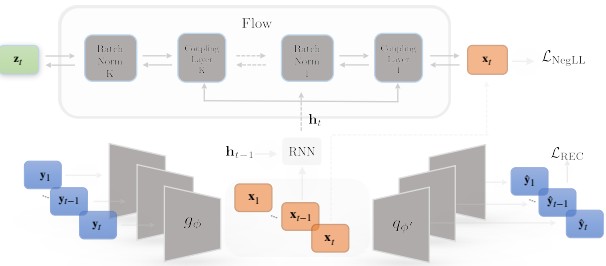

Figure 1: **Information flow during the training process**: The goal of *LatTe-Flows* is to learn a dimensionality reducing mapping (and its approximate inverse) that will produce representations more suitable to dynamics estimation and planning.

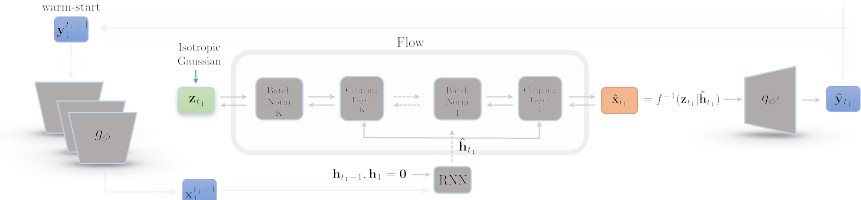

Figure 2: **Forecasting under *LatTe-Flows***: Given a starting state, the flow generates estimates for next time step by sampling from $(\hat{\mathbf{x}}_{t_1}) \sim p(\mathbf{x}_{t_1}|\mathbf{h}_{t_1})$. This process is repeated till our inference horizon. A latent trajectory is mapped back into the input space through the decoder.

overall optimization problem is

$$\min_{\phi,\phi',\theta} \mathcal{L}(\phi,\phi',\theta) = \frac{1}{|\mathcal{B}|(T+\tau)} \sum_{\mathbf{y}_{1:T+\tau}\in\mathcal{B}} \sum_{t=1}^{T} \mathcal{L}_{\text{REC}}(\phi,\phi') + \lambda\mathcal{L}_{\text{NegLL}}(\theta), \qquad (2)$$

where $\lambda \geq 0$ is a hyperparameter that balances the two losses.

Given a trained model, our goal is to produce a set of future trajectories given the past (see Figure 2). Note that, instead of producing samples directly in a possibly very high dimensional feature space, the generator first produces Monte Carlo samples from the predictive distribution $p(\mathbf{x}_{T+1}, \ldots \mathbf{x}_{T+\tau}|\mathbf{x}_1, \ldots, \mathbf{x}_T)$ in the lower dimensional embedding space, by sequentially sampling from $p(\mathbf{x}_t|\mathbf{h}_t)$, updating $\mathbf{h}_t$, and passing the latent sequences through $q_{\phi'}$ to map $\mathbf{x}_t$ back to the original domain.

## 4 EXPERIMENTAL RESULTS

### 4.1 RESULTS ON PUBLICLY AVAILABLE DATASETS

We use publicly available datasets to evaluate the accuracy of forecasts of future observations $\mathbf{y}_t \in \mathbf{R}^N$ compared to to a wide range of existing baselines (described in the Appendix). The **Solar** dataset, with $N = 137$ and $T_{\text{total}} = 4832$, contains hourly solar power production records. The **Electricity** dataset with $N = 370$ and $T_{\text{total}} = 32303$, is a hourly time-series of the electricity consumption of 370 customers. The **Traffic** dataset, with $N = 963$ and $T_{\text{total}} = 4049$, contains hourly occupancy rates of car lanes. **Wiki**, with $N = 2000$ and $T_{\text{total}} = 792$, is a daily time-series daily page views of Wikipedia articles. **Taxi**, with $N = 1214$ and $T_{\text{total}} = 1488$, is a spatio-temporal traffic time-series of New York taxi rides taken every 30 minutes.

Batch sets are formed by randomly sampling context and adjoining prediction horizon interval sized windows from the complete time-series history of the training data. For the test set, we perform rolling prediction evaluation. Following (Salinas et al., 2019a), **Solar**, **Electricity**, and **Traffic**, accuracy is measured on 7 rolling time windows, for **Traffic** we use 5 time windows, and for **Taxi** 57 windows are used in order to cover the full test set. We perform rolling prediction evaluation:

| Data set | Solar | Electricity | Traffic | Taxi | Wikipedia |
|---|---|---|---|---|---|
| VRNN (Chung et al., 2015) | **0.133**±0.009 | 0.051±0.001 | 0.181±0.002 | 0.139±0.015 | 3.400±0.322 |
| KVAE (Fraccaro et al., 2017) | 0.344±0.002 | 0.051±0.025 | — | — | 0.099±0.012 |
| TCNF LSTM - RealNVP (Rasul et al., 2020) | 0.365±0.002 | 0.027±0.001 | **0.026**±0.001 | 0.182±0.013 | 0.097±0.024 |
| TCNF LSTM - MAF (Rasul et al., 2020) | 0.378±0.032 | 0.025±0.002 | **0.028**±0.002 | 0.172±0.001 | 0.097±0.033 |
| Vec-LSTM - ind-scaling (Salinas et al., 2019a) | 0.391±0.017 | 0.025±0.001 | 0.087±0.041 | 0.506±0.005 | 0.133±0.002 |
| Vec-LSTM -lowrank-Copula (Salinas et al., 2019a) | 0.319±0.011 | 0.064±0.008 | 0.103±0.006 | 0.326±0.007 | 0.241±0.033 |
| GP - Scaling (Salinas et al., 2019a) | 0.368±0.012 | **0.022**±0.000 | 0.079±0.000 | 0.183±0.395 | 1.483±1.034 |
| GP - Copula (Salinas et al., 2019a) | 0.337±0.024 | 0.024±0.002 | 0.078±0.002 | 0.208±0.183 | **0.086**±0.004 |
| *LatTe-Flows* - Real NVP | **0.122**±0.00 | 0.022±0.001 | 0.045±0.002 | **0.128**±0.015 | 0.106±0.012 |
| *LatTe-Flows* - MAF | 0.217±0.001 | **0.020**±0.001 | 0.056±0.002 | **0.131**±0.012 | **0.096**±0.017 |

Table 1: Test set CRPS comparison (lower is better) of models from V-RNN, GP Copula, TCNF, and our model variants LatTe with Real-NVP, LatTe with-MAF. The mean and standard errors are obtained by re-running each method three times with random initializations. Best performance is highlighted in bold.

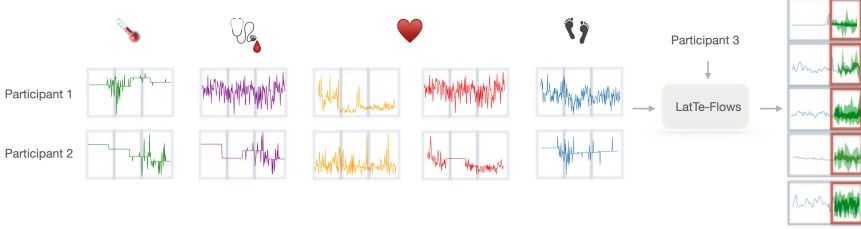

Figure 3: *LatTe-Flows* **models the joint trajectory of vital signals collected from wearables**: The model is trained to produce 20-day forecasts of vital signals, given 30-day multivariate observations of daily heart, blood pressure, and physical activity measurements, collected by wearable devices and manual logging from 1630 healthy individuals. Each participant's set of sensor measurements is segmented into 50 day windows, further subdivided into two parts: the context window (30 days) and the prediction window (20 days). Once the model is trained, given at least 30 past days of a specified participant in the testing set, we produce multivariate forecasts that capture changes over the vital signals through time.

24 time-points per window, last 7 windows for testing for traffic and electricity, and 14 per window with last 4 windows for wiki.

We evaluate forecasting performance starting on equally spaced time points after the last observed training point. Since the marginal continuous ranked probability score (CRPS) cannot assess whether dependencies across time-series are accurately captured, we report the the *Continuous Ranked Probability Score* (CRPS) (CRPS-Sum). This metric is obtained by first summing across the $N$ (ground-truth data and forecasted) time-series, which yields a CDF estimate $\hat{F}_{\text{sum}}(t)$ for each time point. The results are then averaged over the prediction horizon. We take 100 samples to estimate the empirical CDF in practice. Table 2 lists the mean CRPS-Sum values averaged over 10 independent runs with standard deviations and shows that the model sets the new state-of-the-art on most of the benchmark data sets.

## 4.2 RESULTS ON AH&MS DATASET

Biometrics collected from wearable devices, including heart rate measurements and activity levels (e.g., step counts, standing hours) throughout the day, are rich sources of information that can yield crucial insights into the health trajectory of an individual. Vital signals from wearables show feasibility for accurate prediction of clinical laboratory measurements (Dunn et al., 2021). These measurements also show promise to detect acute infections—cosinor models fit to diurnal heart rate variability (HRV) patterns can detect pre-symptomatic COVID-19 infection (Hirten et al., 2021), and anomaly detection algorithms using resting heart rate (RHR) and step counts can identify pre-symptomatic COVID-19 (Alavi et al., 2021). Biometrics also show sensitivity to detect common colds (H1N1 and rhinovirus) (Grzesiak et al., 2021). Beyond acute infections, wearable measurements can provide insight into cardiovascular health, as higher RHR is significantly associated with coronary artery disease, stroke, and sudden death (Zhang et al., 2016). All-cause and cardiovascu-

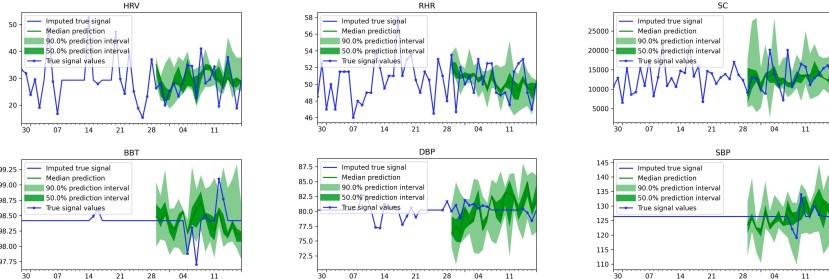

Figure 4: Qualitative predictions for the 20 day forecast window produced by *LatTe-Flows* for AH&MS signal observations (HRV, RHR, SC, BBT, DBP, SBP) of an individual. We estimate 10 trajectories for a latent dimension of $D = 5$ and plot the estimates. Note that the forecast is displayed in terms of a probability distribution: the shaded areas represent the 50% and 90% prediction intervals, respectively, centered around the median (dark green line). The ground truth is overlaid in blue with stars denoting true observations and linear interpolation between the points.

| | NMSE-BBT | NMSE-BPD | NMSE-BPS | NMSE-HRV | NMSE-RHR | NMSE-SC | CRPS-Sum |
|---|---|---|---|---|---|---|---|
| LatTe, D=3 | $0.0124 \pm 0.0028$ | $0.0038 \pm 0.0021$ | $0.0117 \pm 0.0464$ | $0.0828 \pm 0.0718$ | $0.0145 \pm 0.0115$ | $0.3650 \pm 0.4169$ | $0.1068 \pm 0.0154$ |
| LatTe, D=4 | $0.0114 \pm 0.0027$ | $0.0033 \pm 0.0018$ | $0.0115 \pm 0.0040$ | $\mathbf{0.0593 \pm 0.0650}$ | $0.0146 \pm 0.0117$ | $0.3264 \pm 0.4315$ | $0.1027 \pm 0.0156$ |
| LatTe, D=5 | $\mathbf{0.0047 \pm 0.0001}$ | $\mathbf{0.0015 \pm 0.0011}$ | $\mathbf{0.0089 \pm 0.0006}$ | $0.0686 \pm 0.0071$ | $0.0073 \pm 0.0053$ | $\mathbf{0.3080 \pm 0.3061}$ | $\mathbf{0.0841 \pm 0.0332}$ |
| DeepVAR | $0.0277 \pm 0.0021$ | $0.0092 \pm 0.0033$ | $0.0107 \pm 0.0006$ | $0.0817 \pm 0.0056$ | $0.0129 \pm 0.0148$ | $0.4129 \pm 0.4823$ | $0.1281 \pm 0.0154$ |
| GP - Copula | $0.0223 \pm 0.0016$ | $0.0062 \pm 0.0043$ | $0.0105 \pm 0.0056$ | $0.1015 \pm 0.0516$ | $0.0427 \pm 0.0536$ | $0.3206 \pm 0.4152$ | $0.1208 \pm 0.0181$ |
| TCNF LSTM-MAF | $0.0225 \pm 0.0033$ | $0.0081 \pm 0.0031$ | $0.0116 \pm 0.005$ | $0.1165 \pm 0.0593$ | $0.0718 \pm 0.0664$ | $0.3208 \pm 0.3246$ | $0.1052 \pm 0.0166$ |
| TCNF Transformer-MAF | $0.0224 \pm 0.0018$ | $0.0089 \pm 0.0024$ | $0.0109 \pm 0.049$ | $0.1094 \pm 0.0539$ | $\mathbf{0.0064 \pm 0.0226}$ | $0.3144 \pm 0.2726$ | $0.0925 \pm 0.0155$ |

Table 2: We present the mean and standard errors obtained by Deep-VAR, GP Copula, TCNF, and our model variants LatTe with Real-NVP. The evaluation criterion is the Test set Normalized Mean Square Error (NMSE) comparison (lower is better). Best performance for each column is highlighted in bold.

lar mortality is also indicated by lower HRV (Singh et al., 2018). HRV is also found to indicate dysregulation of the autonomic nervous system, and thus is implicated in the development of hypertension (Schroeder et al., 2003). Joint modeling of vital signs from wearable devices continues to yield powerful signals for early detection of disease and health decline, as well as indicate general well-being and fitness. The relatively low cost of wearables and longitudinal nature of measurements compared to in-clinic evaluations motivates continued research in this area.

To this target, we apply *LatTe-Flows* for multivariate signal forecasting: given the past 30-day measurements of a group of vital sensor signals as input, we wish to predict the future 20-day values of the multivariate input time-series. We evaluate the empircal performance of our method on the Apple Heart and Movement Study (AH&MS) dataset. The AH&MS study was sponsored by Apple and conducted in collaboration with the American Heart Association and Brigham and Women's Hospital. The study was approved by Advarra IRB and data were collected in accordance with the IRB approved consent form.

**AH&MS dataset description**: The dataset contains signals collected in real-world environments, which constitute a measure of cardiovascular and autonomic nervous system activity as well as movement and activity metrics. For the participants enrolled in this study, this dataset consists of passively collected sensor signal data from wearable devices in addition to self-reported measurements. This rich collection of signals comprises a partial view of individual biometric information as it evolves over time (see Figure 3).

Our data consists of passively collected observations: Resting Heart Rate (**RHR**), Heart Rate Variability (**HRV**), Step Count (**SC**), and user logged information such as Diastolic Blood Pressure (**DBP**), Systolic Blood Pressure (**SBP**), and Basal Body Temperature (**BBT**). HRV is calculated in the time-domain as the standard deviation between heartbeat measurements (defined by measured R-R intervals). Each time-series measurement is aggregated at the daily level and averaged if there are multiple measurements per day (for RHR, HRV, DBP, SBP, BBT) or summed (for SC).

**AH&MS dataset analysis and results**: Given a training set comprised of multiple context windows of 30 daily observations of BBT, DBP, SBP, HRV, RHR, and SC measurements of 1630 individuals,

the goal is to jointly learn each signal output for the next 20 days, by modeling the temporal dependencies of latent representations of the signals while also learning a meaningful lower-dimensional space. We project another signal captured in our dataset, $VO2_{max}$, (omitted from training) onto the learned latent space to show that our model captures meaningful representations(Figure 5).

Using the Normalized Mean Square Error (NMSE) on each individual time-series, and CRPS-Sum as an evaluation metric, we qualitatively assess test-time signal predictions produced by our method for different latent-space dimensionality, $D = \{3, 4, 5\}$. We compare our approach against existing classical multivariate methods described in the previous section. Because of different scales in each signal, we first normalize each signal by the sum of its absolute values before computing this metric. The results are reported in Table 2, where in average, *LatTe-Flows* with $D = 5$ achieves the best signal predictions.

In Figure 4, we demonstrate the quality of predictions produced by the proposed model—we show the predicted median, $50\%$ and $90\%$ distribution intervals of for HRV, RHR, SC, BBT, DBP, and SBP in the future 20 day window over a randomly chosen individual. In this case, the reduced dimensionality is 4 (see more forecasting results in the Appendix). Interpretable representation learning on time-series is a fundamental problem for uncovering the latent structure in complex systems, such as sensor streams of vital signals. To reveal the underlying factors controlling accurate forecasts, we train the model with $D = 2$ and $D = 3$, and visualize the representations across 50 time points

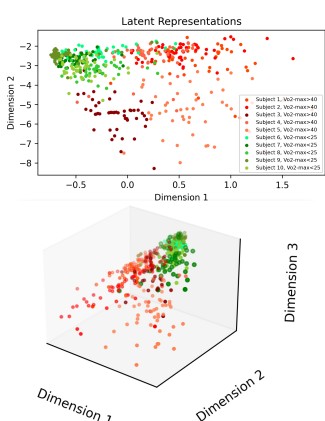

Figure 5: **The latent manifold of the data embedded in 2D latent space (left) and 3D latent space (right) learned by the auto-encoder.** Each point represents a latent signals' representation for a specific time point. Every different color represents a different individual, but those with lower VO$_2$max values are represented in red tones, while those with higher VO$_2$max values in green tones. We observe that learned maps from *LatTe-Flows* clusters subjects according to their estimated fitness level.

for eight different individuals. These individuals belong to two disjoint cardiorespiratory fitness levels: the first group consists of four subjects with low values of estimated VO$_2$max (Apple, 2021), VO$_2$max $< 25$, and the second group consists of four subjects with high values of VO$_2$max, VO$_2$max $> 40$. Maximum oxygen consumption (VO$_2$max) is the measurement of the maximum amount of oxygen a person can utilize during intense exercise and is considered the gold standard for determining an individual's cardiorespiratory fitness and performance capacity. The latent representations for both $D = 2$ and $D = 3$ indicate that the mapping learned by the optimization leads to a space with easily visualized clusters that well-separates individuals according to VO$_2$max (Figure 5). Our latent space estimated using RHR, HRV, BBT, DBP, SBP, and SC corroborates other recent methods that use step count, sedentary time, and moderate-vigorous physical activity to associate with VO$_2$max (Nayor et al., 2021).

## 5 DISCUSSION

In this paper, we developed a general tool for multivariate forecasting that scales to large collections of dependent time-series. We have shown that the combination of representation learning, flexible density models and auto-regressive structures can produce accurate forecasts by modeling dynamics on a low-dimensional subspace. Our model is competitive with and often outperforms state-of-the-art time-series forecasting methods on benchmark applications on publicly available datasets. We further analyzed a challenging practical application of predicting sensor signal trajectories over time. While here our focus is on time-series forecasting, future work will extend our framework to many tasks that require learning over high-dimensional time-series: imputation, interpolation, anomaly and out of distribution detection, and general sequence generation.

## 6 ETHICS STATEMENT

We study the joint trajectory of biometrics over time, a domain that has seen success across health applications. With this success and subsequent increase in research comes an increased need for model, algorithm, and data validation. We acknowledge that any algorithm deployed in a healthcare setting should be subject to validation efforts. We analyze data from the Apple Heart and Movement Study, a study containing information from human participants. The Apple Heart and Movement Study is a collaboration between Apple and Brigham and Women's hospital to understand the factors underlying physical health. Participants in the study must sign an Advarra IRB approved informed consent form before being able to participate in the study. Participation in the study is entirely voluntary and without compensation.

## 7 REPRODUCIBILITY STATEMENT

In the Appendix we provide detailed information and description on the architecture and hyperparameters to reproduce the experiments for each publicly available dataset considered. Although a significant part of this paper considers the Apple Heart and Movement Study, which is not publicly accessible, we provide necessary insights and perform various experiments to showcase the applicability of the proposed model on practical healthcare-related tasks (see details in the Appendix).

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
