# OpenReview forum: "LatTe Flows: Latent Temporal Flows for Multivariate Sequence Analysis "
_ICLR.cc/2022/Conference — ICLR 2022 Submitted_

### Official Review · Reviewer_kY2T · 2021-11-02

**Correctness:** 2
**Technical Novelty And Significance:** 1
**Empirical Novelty And Significance:** 1
**Recommendation:** 1
**Confidence:** 5

**Main Review:**

I have to admit that the paper is a boring read. The combination of some rudimentary ideas such as learning some framewise latent space representations (ideally via some flow model) and then using some sort of RNN to capture temporal dynamics within is far from seemingly  novel, let alone interesting.

The experimental results are unimpressive; performance differences are marginal, comparisons are not against the SOTA in the field, and the considered datasets are far from honestly high-dimensional; I would like to see tens of thousands of dimensions to be convinced.

**Summary Of The Paper:**

In essense, the proposed method leans some lower-dimensional representation of the frames pertaining to observed sequences, and then postulates an RNN-type of model in the leaned latent space to model temporal dynamics. The whole pipeline is trained end-to-end.

**Summary Of The Review:**

The method is trivial. The experimental results do not support publication of an methodologically boring paper.

---

> ### Author Response · Authors · 2021-11-23
> **Response to Comments by Reviewer kY2T**
>
> We  would  like  to  separately  address  the  two  major  points  made  by  Reviewer  kY2T.
>
> We  will begin  with  issues  regarding  the  novelty  of  the  paper.   Sequential  higher-level  latent  variables  models  have been employed before in many practical applications such as audio and video modeling, enabling time-series models  to  capture  complex  dynamics  and  improve  prediction  performance.   Previously  proposed  methods are fundamentally different from our approach, since our model is not trained with a variational bound but instead  optimizes  a  balanced  term  comprised  with  2  different  kinds  of  losses:   the  first  involving  explicit latent distribution modeling of temporal transitions and the second involving identification of deterministic dimensionality reducing/expanding functions.  The proposed cost function is derived from the observation that the  conditional  expectation $E[y_{T+1}, \cdots, y_{T+\tau}| y_{1:T}]$ can  be  approximated  by  first  sampling  a  “compressed” vector$x_t\in \mathbb{R}^D$ from the latent distribution of temporal transitions p(·|x_{1:T}) for a future horizon of τ values, and then applying a dimensionality expanding mapping q, to each these vector representations in order to translate them back in $\mathbb{R}^N$.  According to our experimental analysis (please see details in the supplementary material), using the proposed model one can predict future trajectories by compressing the observed sequence in a much lower-dimensional space.  Thus, one can significantly reduce the complexity and size of the temporal model, since the probabilistic model is applied on a much lower dimensional space.  Last but not least, although in our experiments we use normalizing flows, one could easily replace themwith other expressive density estimation models that enable point-wise PDF estimation.  To the best of our knowledge, our work is the first to use flows to explicitly model latent sequences and jointly learn a decoder mapping, to translate predictions in the original N−dimensional space.
>
> We have clearly explained why we target latent forecasting and we experimentally verify the benefits of the proposed method on both widely used public datasets (note that we chose these standard datasets for easier comparison) and a novel application based on vital sensor signals from the AHMS dataset against 8 SOTA (mainly) probabilistic forecasting methods from the literature to show the performance improvement using the proposed alternative training criterion.  Although, vital sensor signal predictions using the proposed method are significantly better, an improved performance (which is not at all marginal – see prediction error on Solar or Taxi) is noticed over the real-data as well.  Keep in mind that Traffic dataset has dimensionality N= 963, Taxi has dimensionality N=1214, and Wiki has dimensionality N=2000.  For density estimation, at least, those scales are perceived as high dimensional data.

---

### Official Review · Reviewer_s1RF · 2021-11-03

**Correctness:** 3
**Technical Novelty And Significance:** 2
**Empirical Novelty And Significance:** 3
**Recommendation:** 5
**Confidence:** 4

**Main Review:**

Though the idea is very interesting, and the experimental results are also very encouraging, I have below major/minor concerns on this paper:

-- Major 1:
The idea of modeling the latent representations with normalizing flow is not new. Even sequential normalizing flow has been proposed recently (see [1] and [2]). It is not clear to me what is the major novelty in terms of methodology.

-- Major 2:
Some descriptions are not clear to me. For example, in section 3.2, p_{theta}(y|h) and p_{theta}(x|h) should be distinguished. According to Figure 1, it seems that h_t would affect x_t, but not the whole flow process. Are all the revertible transformations between the low-dimensional sample z_t and x_t included in the parameter set theta? Also, how your “NegLL” term affects the normalizing flow is not illustrated in the draft.

--Minor concerns:
1. The model does not improve in every scenario, but the overall performance does look good.
2. Table two is referred to twice in the draft, but table one is not referred mentioned in the main text. I guess one reference should be corresponding to table one?
3. In Table two, what are the latent dimensions of all the other methods? Did the authors also tune on different approaches?
4. How does parameter lambda affect the learned representation and the reconstruction?
5. The authors do not compare to TLAE, though you claimed this is (one of) the most related work.
6. The authors mention that TLAE is using diagonal Gaussian prior, which is restrictive. According to Figures one and two, it seems the proposed method is also using a restrictive prior?
7. It would be good to provide additional experiments to show that your methods scale well rather than on performance.
8. There is a typo in line 9 of the page 4.

[1]: https://arxiv.org/pdf/2010.03172.pdf
[2]: https://arxiv.org/pdf/1901.10548.pdf


**Summary Of The Paper:**

The authors propose a novel approach that scales well for multivariate sequence forecasting tasks. They offer to use conditional normalizing flow to capture the sequence dynamics in the latent space and then use auto-regressive architecture to decode the signal in the original space. The authors clearly explain why they prefer latent forecasting and experimentally verify the benefits of the proposed method on both widely used public datasets and the AH&MS dataset.


**Summary Of The Review:**

Right now, I am more inclined not to accept the paper. However, I am happy to discuss and also change my evaluation if I missed anything important.

---

> ### Author Response · Authors · 2021-11-23
> **Response to Comments by Reviewer s1RF**
>
> " The idea of modeling the latent representations with normalizing flow is not new.."
>
> Response: Please see the clarification addressed to Reviewer f9pX for a detailed answer regarding the novelty of our paper.There are major differences between the proposed method and [1], [2].  In [1] an auto-regressive latent variable model optimized with variational inference is extended with an auto-regressive flow that further transforms the output of the latent variable model while allowing to compute exact conditional probability.  This is fundamentally different from our approach, since our model is not trained with a variational bound but instead optimizes a balanced term comprised with 2 different kinds of losses:  the first involving explicit latent distribution modeling of temporal transitions and the second involving identification of deterministic dimensionality reducing/expanding functions.  Again with respect to [2], the difference lies within the approach of the optimization method, since they also employ a variational inference and training, as well the type of data to be modeled (for [2] discrete random variables such as text).  We acknowledge the capabilities of latent representations with normalizing flows for time series analysis, the experimental analysis and application employed in our work further strengthens this approach but also provides an alternative formulation to variational training.
>
> "Some descriptions are not clear to me..."
>
> Response: Thank you for noticing that -- Indeed $L_{\text{Negll}}$ is defined on the latent vector $x_t$, so $p_{\theta}(x_t|h)$ should have been used. We will correct the typo in Section 3.2.
>
> Yes.  The temporal component, modeled with an RNN, outputs a representation of compressed informationof previous timesht, which is then used as the condition in the flow transformations.  We concatenate $h_t$ to the inputs of the scaling and translation function approximators of the coupling layers, and where θ denotes the set of all parameters of both the flow and the RNN (see details in Rasul et. al.).  Combined with the dimensionality expansion functions (learned through the decoder), the architecture allows for trajectories to be sampled for vector predictions into the future.  Notice that all components in the architecture are learned jointly to maximize the prediction accuracy. We combine two objectives in our cost function:  one for reconstruction and one for forecasting (“NegLL”,which is controlled by the regularization parameter λ).  Across all experiments, we choose the regularization parameter λ to make sure that the reconstruction and regularization losses in the objective have similar order of magnitude, and report each selected value in the supplementary material.  Potentially, a more thorough hyper-parameter search is going to improve performance further.
>
> "The model does not improve in every scenario, but the overall performance does look good."
>
> Response:   Our method achieves an improved performance on most publicly available datasets, and significantly better predictions of vital signals with respect to the baselines considered.  Notice that, since the probabilistic model is applied in a much low-dimensional space, in addition to an improved performance, our method achieves a substantial reduction in the complexity and size of the temporal model.  This reduction depends on the size of latent dimensionality – A detailed analysis for this hyper-parameter for each dataset is provided in the supplementary material.
>
>
> "Table two is referred to twice in the draft, but table one is not referred mentioned in the main text.  I guess one reference should be corresponding to table one."
>
> Response: Thank you for noticing that.  Indeed Table 1 should be referenced instead in Section 4.1.  We will correct the reference in our revised manuscript.
>
> "In Table two, what are the latent dimensions of all the other methods?  Did the authors also tune on different approaches?"
>
> Response: We only perform dimensionality reduction for our method and report the performance for different values of D.  Regarding the baselines, the tunable parameters are separately tuned to optimize the performance of each method considered.  – Table 2.  reports the best performance for all baselines from our experimental section.
>
> "The authors mention that TLAE is using diagonal Gaussian prior, which is restrictive. According to Figures one and two, it seems the proposed method is also using a restrictive prior?"
>
> Response:   TLAE depends primarily on the encoder’s ability to capture complex dependencies among series andintroduces fairly simple probabilistic structure on the latent variables.  We do not impose any prior distribution –Instead, we let our model explicitly learn the latent temporal distribution by a flexible family of probabilisticmodels (normalizing flows).  Intuitively, the expressiveness for our model comes from both the encoder architecture and the latent temporal model.

---

### Official Review · Reviewer_f9pX · 2021-11-09

**Correctness:** 3
**Technical Novelty And Significance:** 1
**Empirical Novelty And Significance:** 2
**Recommendation:** 3
**Confidence:** 3

**Main Review:**

**Strengths**
This paper has done good comparative evaluations of the proposed framework against baselines to demonstrate that it significantly outperforms the state-of-the-art on multi-step forecasting benchmarks.


**Weaknesses**
1. I'm not convinced that the proposed latent variable estimation framework is completely novel. As pointed out by the authors, the core component of the proposed framework is MAF autoregressive flow model, which has been applied for many sequential tasks already.

2. $L_{\text{Negll}}$ in Eq (1) is different from the definition in Section 3.1. One is defined on observed variables y and the other on latent variables x. It'll be better if the authors derive Eq (1) from a pure MLE perspective, instead of combining reconstruction loss with flow loss. I personally feel the loss function is not theoretically justified because I'm not sure what it is optimizing but I may be wrong.


**Summary Of The Paper:**

This paper proposes LatTe Flows, a novel autoregressive flow model with autoencoders to learn latent embeddings from time series. The authors claim that it can model cross-series dependencies across time while scaling to large dimensions with an end-to-end training process.

**Summary Of The Review:**

I suggest the rejection of this paper in its current version. Although the authors have done comparisons of their framework with baselines, the proposed model is not totally novel.

---

> ### Author Response · Authors · 2021-11-23
> **Response to Comments by Reviewer f9pX**
>
> "Novelty" : First of all, we would like to address the concern that the novelty of our work is limited.  Our work not only proposes a simple yet effective time-series generative model and its application for multivariate forecasting, but also has some important implications. Our main contributions are two fold:
>
> 1.Basically, we make the novel observation that the conditional expectation $E[y_{T+1},···,y_{T+τ}|y_{1:T}]$ can be approximated by first sampling a “compressed” vector $x_t\in \mathbb{R}^D$ from the latent distribution of temporal transitions $p(\cdot|x_{1:T})$ for a future horizon of τ values and then applying a dimensionality expanding mapping q, to each these vector representations in order to translate them back in $\mathbb{R}^N$. On the practical side, our experiments further support the assumption that by projecting a high dimensional multivariate time series on an intermediate (much lower dimensional) latent space that preserves most of the essential information (cross-series correlations and dependencies), one can obtain an improved forecasting performance as a result of denoising benefits of modeling compact representations.  The flexibility of our approach for multivariate forecasting is partly motivated by the fact that we explicitly learn the latent space structure through an expressive time-conditioned distribution model of temporal transitions without strong assumptions of traditional multivariate models (low-rankness, independent components).  Although in our experiments we use normalizing flows, one could easily replace them with other expressive density estimation models that enable point-wise PDF estimation.
>
> 2.Probabilistic forecasting of high dimensional multivariate time series is a notoriously challenging task interms of computational burden.  The approach is intuitively beneficial in that the (deterministic) encoding function can significantly reduce the dimensionality of the input vectors and thus reduce the complexity and size of the temporal model (without DR, RNN introduces quadratic complexity in the number of series, whereas in our case it can be much lower, depending on the latent dimensionality).  In case of the Electricity dataset, for example, the input dimensionality N= 370 can be reduced to a latent dimensionality D= 32, incurring a percentage of 91.4% dimensionality reduction, and thus indicating the scalability of the proposed method.
>
> Additionally part of the novelty includes the proposed application:  we show that LatTe-Flows can accurately model the joint trajectory of vital signals collected from wearables.  Other than improved forecasting performanceon the AH&MS dataset, we visualized the 2-dimensional and 3-dimensional learned latent representations fromour model to showcase interpretable low-dimensional embeddings.  These emebeddings show that LatTe-Flows is able to easily identify participants’ VO2max, a main indicator and summary of cardiorespiratory fitness, while only being trained on relatively lower-level vital signals like resting heart rate and heart rate variability.
>
> " $L_{\text{Negll}}$ Eq (1) is different from the definition in Section 3.1.  One is defined onobserved variables y and the other on latent variables x.  It’ll be better if theauthors derive Eq (1) from a pure MLE perspective, instead of combining reconstructionloss with flow loss.  I personally feel the loss function is not theoretically justified because I’m not sure what it is optimizing but I may be wrong."
>
> Thank you for noticing that – Indeed $L_{\text{Negll}}$ is defined on the latent vector $x_t$.  We will correct the typo in Section 3.2.
>
> One can not derive Eq (1) from a pure MLE perspective and the reason is the following.  The multivariate extension of change-of-variables theorem suggests that given random vectors $y_t$ and $x_t$ which are related by a mapping $y_{t} = q(x_{t})$ one can transform the PDF of random vector $x_t$ into the PDF of random vector $y_t$, with the requirement that the transformation q is a bijection.  In our case, the dimensionality reducing mapping: $q: R^D \rightarrow R^N$, clearly cannot satisfy this constraint since bijective transformations are required to preserve dimensionality.
>
> This restriction inspired us to formulate our current loss function which is indeed theoretically justified using  the  following  argument:  If  the  latent  distribution  of  temporal  transitions $p(\cdot|x_{1:T})$  can  be  recovered by an expressive family of time-conditioned distributions (such as the method recently proposed in Rasul et al.), we can perform forecasting by sampling from the latent conditional distribution and then apply a (deterministic) mapping function q to transform (sampled) vector realizations to the original input space.  This is the intuition behind the loss function:  the reconstruction term is used to learn the mapping function q and the NLL to recover the temporal distribution of latent vectors.

---

> > ### Comment · Reviewer_f9pX · 2021-11-29
> > **Thanks for the response**
> >
> > Thank you for your detailed response. I've read all the other reviewers' comments and it seems that some of us agree that the comparative experiments are extensive and the experimental results are very encouraging. However, we also seem to agree that the proposed method, which combines encoder/decoder structure with autoregressive flows on the latent space, is not novel enough in its current version:
> >
> > 1. Modeling high-dimensional time series on the latent space is not considered novel. For instance, most video prediction studies nowadays roll out the temporal dynamics on the latent space;
> >
> > 2. I'm still not convinced that combining encoder/decoder architecture with the flow model is better than using flow directly. For example, in this [paper](https://arxiv.org/pdf/1903.01434.pdf), a flow model is directly used for the high-dimensional video prediction task by modeling the temporal dynamics on the latent space. The benefits of using flow are that MLE, instead of its lower bound, can be optimized directly. If the authors believe flow needs a lot more layers to achieve the expressive power of a non-invertible encoder, why not use temporal VAE directly, of which at least the ELBO objective function is justified.
> >
> > I hope the authors do not take my comment too negatively, but given all the comments, I retain my original score.

---

### Decision · Program_Chairs · 2022-01-20

**Decision:**

Reject

**Comment:**

This paper proposes a new autoregressive flow model with autoencoders to learn latent embeddings from time series. The authors conducted extensive comparative experiments, and the experimental results are very encouraging. However, the proposed method, as a combination of the encoder/decoder structure and autoregressive flows on the latent space, does not seem novel enough.